# Serial histological changes in the cartilaginous eustachian tube in the rat following balloon dilation

**Yehree Kim**[1], **Jeon Min Kang**[2], **Dae Sung Ryu**[2], **Jung-Hoon Park**[2], **Woo Seok Kang**[1], **Hong Ju Park**[1]*

1 Department of Otorhinolaryngology-Head & Neck Surgery, Asan Medical Center, University of Ulsan College of Medicine, Seoul, Republic of Korea, 2 Biomedical Engineering Research Center, Asan Institute for Life Sciences, Asan Medical Center, Seoul, Republic of Korea

* dzness@amc.seoul.kr

**Data Availability Statement:** All relevant data are within the manuscript and its Supporting Information files.

## Abstract

Although balloon dilation has shown promising results in the treatment of dilatory Eustachian tube (ET) dysfunction, the histological effects of ET balloon dilation (ETBD) is unknown because histological examination of the whole human cartilaginous ET is impossible. Animal studies are needed to elucidate the effect of ETBD so we evaluated the histological changes after ETBD in a rat model. The left ET of 20 Wistar rats was dilated with a balloon catheter and the right ET was used as a control. Five rats were sacrificed immediately after ETBD, at 1, 4 and 12 weeks after the procedure for histological examination. The epithelial cells, presence of epithelial hyperplasia, and the proportion of the goblet cells in the epithelium; the vascular structures and dimensions of the submucosa; and presence of cartilage fracture and the area of the ET lumen were evaluated and compared between the groups. Desquamation of nearly all epithelial cells and the fracture of tubal cartilages were observed immediately after ETBD. At 1-week post-ETBD, the ciliated epithelial cells started to recover with epithelial hyperplasia. The goblet cells recovered by 4 weeks post-ETBD and epithelial hyperplasia decreased but was still present at 12 weeks post-ETBD. The depth of the submucosa increased and neovascularization in this region was observed at 1-week post-ETBD and persisted up to 12 weeks post-ETBD. The lumen of the cartilaginous ET increased immediately after ETBD but decreased at 1-week post-ETBD. The cartilaginous ET lumen recovered to the normal value at 4 weeks post-ETBD. This study is the first to describe the serial histological changes to the cartilaginous ET after ETBD and helps our understanding of the histological changes that occur after an ETBD intervention for intractable ET dysfunction.

## Introduction

As the sole connector of the middle ear to the nasopharynx, a functioning eustachian tube (ET) is important for maintaining a healthy well-aerated middle ear [1]. The functions of the ET include middle ear ventilation, as well as the transport and secretion of pathogens and the

**Funding:** This work was supported by the National Research Foundation of Korea (NRF) grant funded by the Korean government (Ministry of Science and ICT) (2020R1F1A1049412 to Park H.J.). The funders had no role in study design, data collection and analysis, decision to publish, or preparation of the manuscript.

**Competing interests:** The authors have declared that no competing interests exist.

nasopharyngeal reflux [2, 3]. A dysfunctional ET can lead to acute and chronic otitis media, one of the most common disease entities encountered in otolaryngology [4]. Conditions that interrupt the proper opening of the ET, such as an inflammatory response within its lumen caused by irritants and infectious reactions, can result in ET dysfunction (ETD). The inability of the ET to open also results in a gas equilibrium failure between the middle ear and nasopharynx. Atmospheric gases diffuse across the venous capillary cell membranes in the middle ear. So dilatory dysfunction of the ET results in a net negative pressure within the middle ear, leading to symptoms such as aural fullness or 'popping sounds', reduced hearing, tinnitus, autophony, otalgia, and imbalance [5].

Until recent years, the understanding of ETD among otolaryngologists was limited, and few treatment options were available [6–11]. Surgical management is now available after the introduction of the ET balloon dilation (ETBD) procedure [12]. Since Ockermann et al. reported their first experience with ETBD in 2010 [13], many studies have since reported this procedure to be feasible and safe for the treatment of ETD [14–21]. The reported success rates of ETBD range from 36–80% [16, 21–24].

Although ETBD is reported to be superior to the conventional medical management of ETD, there is still a population of affected patients who do not respond to this intervention. No consensus has yet been reached regarding the further management of ETBD failure cases. This is mainly due to the limited understanding of the mechanisms underlying ETBD due to a lack of histologic studies of the ET after this surgery. A reported finding after ETBD is microtears in the cartilaginous part of the ET [12] and a resulting decrease in mucosal inflammation and reduced biofilm load [25]. A previous histological study by Kivekäs et al. reported that thinning of the mucosa, shearing of the epithelium, and a crush injury to the submucosa, particularly involving lymphocytic infiltrates, were the immediate responses to balloon dilation, and that a healthy pseudocolumnar epithelium and replacement of lymphocytic infiltrates with a thinner layer of fibrous tissue could be observed postoperatively [25]. One drawback of that prior report however was that the ET mucosal specimens were taken from the nasopharyngeal opening of the tube which does not include the true cartilaginous portion.

Histological examination of the whole human cartilaginous ET after ETBD is impossible. Therefore, an animal study is warranted to verify the histological changes of the ET after ETBD. In this study, we aimed to evaluate the serial histological changes of the ET epithelium, submucosa, and cartilage and the area of the ET lumen immediately after and at 1, 4 and 12 weeks after balloon dilation in a rat model.

## Methods

### Animals

This study was approved by the Institutional Animal Care and Use Committee at Asan Institute for Life Science and conformed to US National Institutes of Health guidelines for humane handling of laboratory animals and is reported in accordance with ARRIVE guidelines (Animal Research: Reporting of In Vivo Experiments). Twenty male Wistar rats, 10 weeks of age (300–350 g; Orient Bio, Seongnam, Korea), were included in our study. All animals were housed at one per cage in a room with a 12-hour contrast cycle at 24 ± 1˚C with a relative humidity of 55 ± 10%. Standard rodent chow and water were provided *ad libitum*. All animals were acclimated for at least 1 week prior to conducting the experiments. All procedures were carried out under anesthesia induced by means of an intramuscular injection of 50 mg/kg zolazepam and tiletamine (Zoletil 50; Virbac, Carros, France) and 10 mg/kg xylazine (Rompun; Bayer Healthcare, Leverkusen, Germany) and all efforts were made to minimize suffering. The left ET was dilated with a balloon catheter and the right ET was used as a normal control in all

20 rats. The rats were assigned to 4 groups, each consisting of 5 animals, for serial histological examination at 4 different time points: immediately (n = 5), at 1 week (n = 5), 4 weeks (n = 5), and 12 weeks (n = 5) after ETBD. All animals were sacrificed by placing the original housing cage in a carbon dioxide chamber and gradually increasing the concentration of carbon dioxide.

## Fluoroscopic eustachian tube balloon dilation

All ETBD procedures were performed under fluoroscopic guidance by an interventional radiologist and otologist. After disinfection of the left external auditory canal with 0.05% chlorhexidine hydrochloride, the left tympanic membrane was punctured using a micro puncture needle (Cook, Bloomington, IN). The needle was directed 45 degrees anteriorly and in a horizontal plane under fluoroscopic guidance. The tip of the needle was located inside the tympanic cavity and about 0.2 ml of contrast media was injected into this cavity to visualize the tympanic orifice of the ET. The tip of the needle was then relocated close to the tympanic orifice. A 0.014-inch micro guidewire (Transend; Boston Scientific, Fremont, CA) was next gently advanced through the ET and eventually out of the nasal cavity. After removal of the micro puncture needle, a micro balloon catheter (Genoss, Suwon, Korea) was introduced over the guidewire from the nasal side and located at the middle portion of the ET to cover the whole tube structure. The balloon catheter was then inflated at a pressure of 10 atmospheres for 1 minute. During balloon inflation, waist formation and subsequent disappearance of the waist was observed. The micro balloon catheter and micro guidewire were removed after deflation of the balloon.

## Histologic examinations

Surgical exploration was performed immediately after sacrifice. The mandible and anatomical structures anterior to the soft palate were then resected and the nasopharyngeal opening of the ET was identified. The soft tissue structures around the ET were dissected away whilst taking care not to disrupt the anatomy of the tube. Head samples were fixed in 10% neutral buffered formalin for 48 hours, followed by decalcification for 2 weeks (Osteosoft, Merck, Germany). The samples were then coronally sectioned from the nasopharynx to posterior wall of the tympanic cavity at intervals of 200 μm which provided 7 slides per each animal. Paraffin blocks were prepared from the sliced head samples from which 5 μm slices were obtained with an MR2258 microtome (Histoline, Pantigliate, Italy) and placed on slides. The slides were then stained with hematoxylin-eosin (H&E).

Whole slides of the ET for each rat were reviewed. The cartilage surrounding the ET was first identified. The section in which the sectioned ET cartilage was in the form of a 'comma' shape, showing both cartilaginous and nasopharyngeal epithelia, was selected for analysis (Fig 1). Evaluations included an assessment of changes in the epithelial cells, in the dimensions of the submucosa (degree of fibrotic changes), or in the lumen. In the epithelium, the presence of goblet cells, epithelial hyperplasia was assessed. In order to quantify the changes of these aspects of the epithelium, the length of the epithelium exhibiting goblet cells, epithelial hyperplasia were measured and divided by the whole perimeter of the ET lumen in the corresponding slide. Epithelial hyperplasia was also assessed by counting the maximum number of cell layers. These measurements were then compared among the 4 study groups (immediate, and 1, 4, and 12 weeks after ETBD) and the normal group (20 specimens from the non-dilated right ET).

In the submucosa, blood vessels were identified and those under the tubal cartilage were counted and analyzed as numbers per slide. After ETBD, the mucosal breaks caused by the

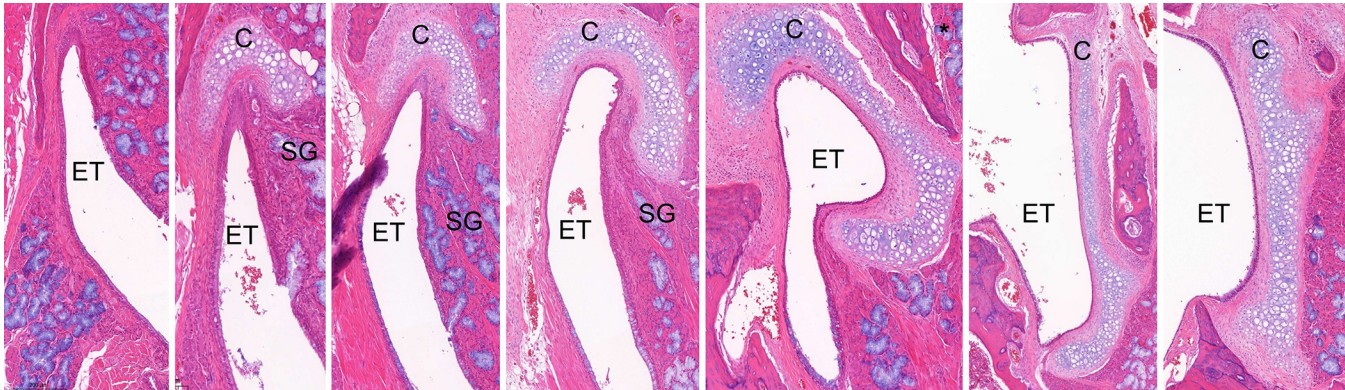

**Fig 1.** Normal histology of the right-side rat ET sectioned from the nasopharynx (left) to the tympanic bulla (right). The tissues were sectioned coronally at 200 μm intervals, in the caudal direction from the medial to lateral sections. C, cartilage; ET, Eustachian tube; SG, submucosal glands (H&E stain). *The section in which the ET cartilage was in the form of a 'comma' shape, i.e. showing both the cartilaginous and nasopharyngeal epithelia, was selected for analysis.

dilatory force of the balloon were replaced by collagen tissue (S1 Fig). To evaluate the extent of fibrosis caused by ETBD, the depth of the submucosa which would correspond to the synthesis of new collagen was measured. The depth of the submucosa of the left ET (balloon) was measured at 9 different points in total in the medial, middle and lateral areas of the surrounding cartilage. The same was done for sections that were 200 μm proximal and distal to the reference slide. The 9 measurements were averaged to obtain the representative depth of the submucosa. The depth of the submucosa of the normal right ET was also measured as a control using the same points measured on the left side.

Due to the variation of the angles at which the ETs were sectioned among the slides, linear measurements alone may not accurately represent the thickness of the submucosa. The area of the submucosa was also therefore calculated. Histomorphometric analysis was conducted using CaseViewer, version 2.4 (3DHISTECH, Budapest, Hungary). The area of the lumen was also recorded. The relative proportions of the observed absolute values were compared.

## Statistical analysis

The data are presented as means ± standard deviations. Statistical analyses were performed using SPSS software (version 24.0; SPSS, IBM, Chicago, IL). When comparing between two samples, significance of data was assessed by Mann-Whitney U test because samples did not pass the normality test. A p-value of less than 0.05 was considered as statistically significant for differences between groups.

## Results

### Histological findings of normal rat eustachian tubes

The epithelium of the rat ET is a ciliated respiratory epithelium. This ET epithelium was divided into cartilaginous and nasopharyngeal ends (Fig 2). The epithelium at the nasopharyngeal end consisted of columnar cells and also contained goblet cells (secretory mucous cells). The epithelium at the cartilaginous end consisted of cuboidal cells and contained few secretory mucous cells. The rat ET was surrounded by cartilage in the posterosuperior direction. A thin submucosa was found between the epithelium and the cartilage. The submucosa consisted of fibroblasts, collagen fibers and a few blood vessels. There was no lymphocytic involvement in

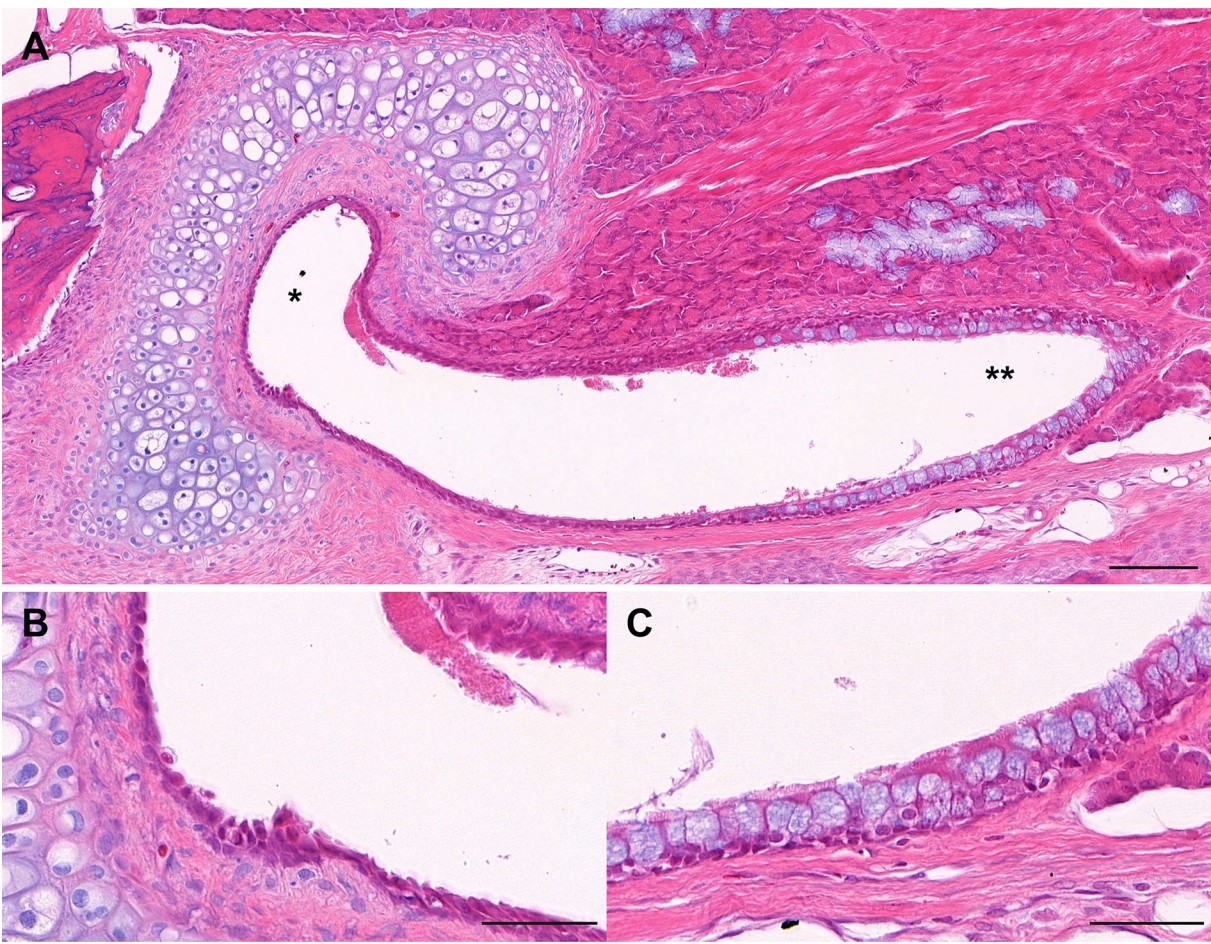

**Fig 2.** (A) Normal histology of the rat Eustachian tube (ET) (15x power field view; H&E stain; scale bar represents 100μm). The ET consists of the cartilaginous end (*) and the nasopharyngeal end (**). (B) High-power field view of the epithelium at the cartilaginous ET (50x power field view; H&E stain; scale bar represents 50μm). (C) High-power field view of the epithelium at the nasopharyngeal end (50x power field view; H&E stain; scale bar represents 50μm).

the rat ET. In addition, as the rat ET extends toward the nasopharynx, the cartilage disappears and submucous glands appear, most notably in the medial portion.

## Changes to the rat eustachian tube epithelium after balloon dilation

All 20 animals completed the whole study. The technical success rate of ETBD was 100%. The epithelial cells at the nasopharyngeal end of the rat ET were destroyed immediately after ETBD. At 1-week post-ETBD, the morphology of these epithelial cells had recovered but goblet cells were not seen. At 4 weeks post-ETBD, goblet cells could be identified but in a less ordered manner. At 12 weeks after the ETBD procedure, the columnar epithelium and goblet cells had fully recovered (Fig 3). At the cartilaginous end of the rat ET, the epithelial cells were desquamated immediately after ETBD. At 1-week post-ETBD, the epithelial cells showed a morphological change to epithelial hyperplasia. At 4 weeks post-procedure, the degree of hyperplasia decreased, and the original morphology was finally recovered at 12 weeks post-ETBD (Fig 4).

Fig 5A shows the changes in the portion of the epithelium with goblet cells in proportion to the whole perimeter of the ET lumen. The goblet cell proportion in the epithelium decreased to zero immediately after ETBD and remained at zero until 1week post-ETBD. At 4 weeks

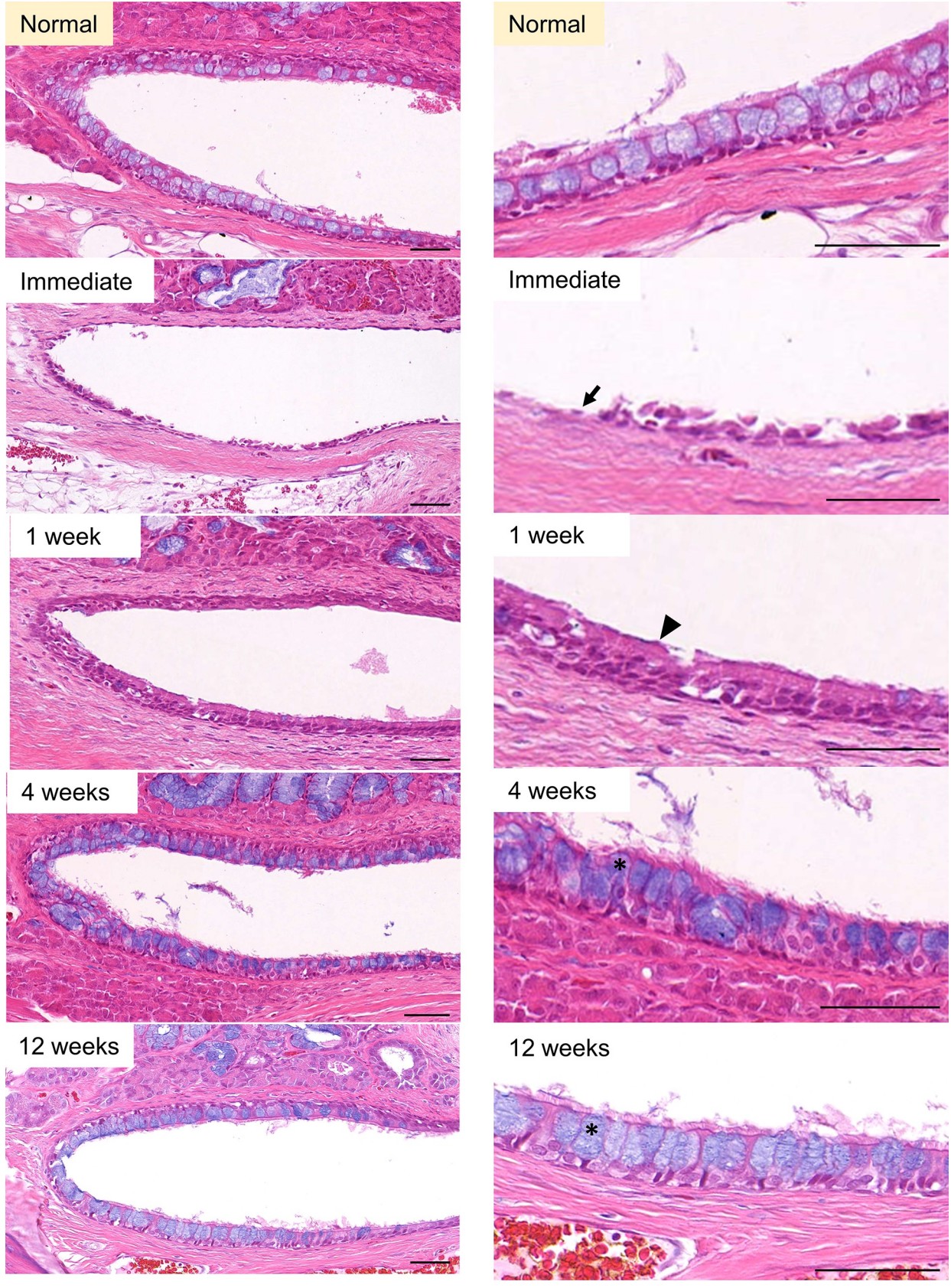

**Fig 3. Serial epithelial changes at the nasopharyngeal end of the rat ET after ETBD.** A 15x power field view and 50x power field sections are shown in the left and right columns, respectively. The epithelial cells were damaged (arrow) immediately after ETBD. At 1-week post-dilation, the morphology of the epithelial cells had recovered but goblet cells were not seen (black arrowhead). At 4- and 12-weeks post-dilation, the columnar epithelium and goblet cells had fully recovered (asterisk). H&E stain, scale bars represent 50μm.

post-surgery however, the proportion of goblet cells increased significantly (28.7 ± 16.5%) to a level that was not statistically different to the normal group (32.0 ± 3.4%). At 12 weeks post-ETBD, the proportion of goblet cells remained at a similar level to the normal group (31.9 ± 6.8%).

Fig 5B shows the changes in the portion of the epithelium showing epithelial hyperplasia in proportion to the whole perimeter of the ET lumen. This level of epithelial hyperplasia significantly increased at 1-week post-dilation (33.7 ± 6.4%), then decreased significantly to 10.1 ± 7.7% at 4-weeks post-ETBD. At 12 weeks after ETBD, the level of epithelial hyperplasia decreased to 4.2 ± 3.7%, which was not statistically different to the normal control. Fig 5C shows the changes of the maximum cell layer count of the epithelium. Immediately after ETBD, all of the epithelia had fallen off and the maximum cell layer count was zero. A significant increase in the maximum cell layer count was evident at 1-week post-dilation (5.0 ± 0.9), which decreased to 2.8 ± 0.4 and 2.4 ± 0.5 at 4 and 12 weeks after the procedure, respectively. The maximum cell layer count was significantly increased at 12 weeks compared to the normal control ($p < 0.05$, Mann-Whitney U test).

## Serial changes in the submucosa after ETBD

Fig 6A shows the changes observed in the vascular structures in the submucosa. Immediately after ETBD, no blood vessels could be identified. The average number of blood vessels per slide was 2.4 ± 1.4, 2.2 ± 1.3 and 2.4 ± 2.0 at 1-, 4-, and 12-weeks post-ETBD, respectively and was greater at 12 weeks compared to the normal control ($p < 0.05$, Mann-Whitney U test). Fig 6B shows the observed changes in the depth of the submucosa. No significant alterations in this depth were evident immediately after ETBD, but it was found to be increased to 157 ±54.3 μm at 1-week post-ETBD. The depth of the submucosa at 12 weeks was 105.2 ± 20.7 μm which was significantly thicker than the normal group (58.5 ± 21.3 μm, $p < 0.05$, Mann-Whitney U test). The submucosa was compared in the different experimental groups by the proportion of the area calculated in each histology section. This value was increased to 23.2 ± 1.3% at 1 week, 24.4 ± 2.6% at 4 weeks, and 25.9 ± 1.8% at 12 weeks after the ETBD procedure and was significantly greater at 12 weeks compared to the normal group (18.7 ± 2.4%, $p < 0.05$, Mann-Whitney U test, Fig 6C).

The proportional area of the ET lumen was compared in each of the histology sections and was found to change throughout the 12-week time period but to reach and maintain a normal level at 12 weeks after the ETBD procedure (Fig 6D).

## Serial changes in the tubal cartilage after ETBD

Out of the 20 rat ETs that were analyzed, 6 showed no frank fracture lines whereas a total of 21 fracture lines were identified in the remaining 14 ETs. These fracture lines were found at 3 distinctive locations: at the midpoint between the medial and lateral lamina of the tubal cartilage, and at the lateral 2 points that divide the lateral lamina into thirds (Fig 7). The frequency at which the fracture lines were identified were 48%, 19%, and 33% respectively, from the midpoint toward the far lateral. Fig 8 summarizes all of the changes identified in this study in the epithelium, submucosa and lumen.

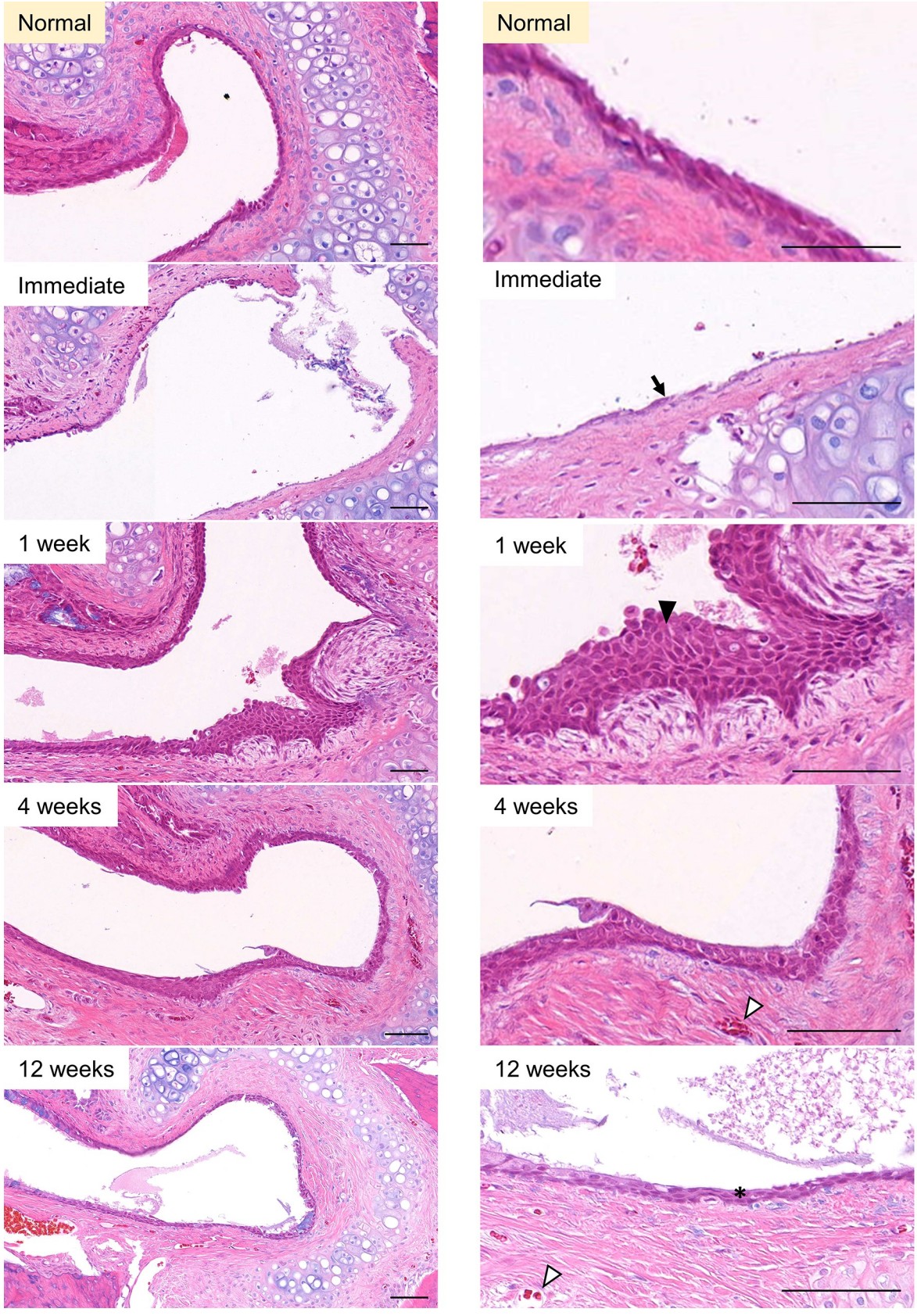

**Fig 4. Serial epithelial changes at the cartilaginous end of the rat ET after ETBD.** A low- and high-power field of view are shown in the left and right columns, respectively. The epithelial cells were damaged (arrow) immediately after ETBD. At 1-week post-dilation, the epithelial cell morphology had changed to epithelial hyperplasia (black arrowhead). At 4 weeks after the procedure, the epithelial hyperplasia had decreased, and recovery to the original morphology (asterisk) was finally observed at 12 weeks post-ETBD. White arrowheads with black borders denote blood vessels. H&E stain, scale bars represent 50μm.

## Discussion

With the introduction of the ETBD procedure, chronic ETD could be surgically managed with a reported success rate of 36–80% [12, 16, 22, 23]. However, 20–64% of ETD patients do not respond to this ETBD treatment modality, thus necessitating further management options. There is no consensus at present about the optimal balloon catheter sizes or lengths to use in patients with ET disorders [26]. The difficulties in developing further management approaches for ETD may lie in the lack of histological studies. Hence, we speculated that a thorough investigation of the histologic changes that occur in the ET after dilation with balloons of different diameters and lengths might provide new insights into treating and managing ETD.

Kivekäs et al. have previously obtained pre- and postoperative biopsy specimens in 13 patients who underwent ETBD [25]. For practical reasons however, histologic studies of the human ET could only be carried out at the nasopharyngeal orifice. The authors suggested in their analyses that the crushing effect of the balloon on lymphocytes and lymphocytic follicles that were later replaced with a thinner fibrous scar is a possible mechanism underlying the therapeutic effects of ETBD [25]. However, the whole length of the ET needed to be investigated to properly evaluate the effects of ETBD, particularly as the cartilaginous ET is considered a crucial structure in maintaining normal ET function. Hence, a proper histological study of the whole ET using appropriate animal models became necessary. Generally, sheep and pigs are considered to be good model systems for the human middle ear [27]. However, a smaller animal model enables larger scale and higher throughput studies, making it more cost-efficient, and more suited to observing the tissue reactions after ETBD which was an important consideration [28]. In our current study, we were able to dilate the ET in the rat using commercially available micro balloon catheters.

We here investigated the serial histological changes in the rat ET after ETBD for up to 12 weeks. Immediately after ETBD, we observed desquamation of nearly all epithelial cells and fracture of the tubal cartilage. At 1-week post-ETBD, the ciliated epithelia cells started to recover via epithelial hyperplasia but goblet cells were still missing. The goblet cells recovered by 4 weeks post-ETBD, but epithelial hyperplasia (maximum cell layer count) was still decreased at 12 weeks post-ETBD. The depth of the submucosa increased and neovascularization in the submucosa was observed at 1-week post-ETBD, which persisted at 12 weeks post-ETBD. The lumen of the cartilaginous ET increased immediately after ETBD but decreased at 1-week post-ETBD due to the increased depth of the submucosa and onset of epithelial hyperplasia. The cartilaginous ET lumen recovered to normal at by 4 weeks post-ETBD.

Increased submucosal fibrosis after ETBD was the significant finding of our current study. The effect of ETBD seems to be related to the fibrosis of the submucosal connective tissue. In terms of prior histologic exams of similar tissues, Modi et al. performed ETBD of the rabbit subglottis [29]. The subglottis and ET both have a cartilaginous framework lined by a respiratory epithelium and mucosa. Thirty days after dilation, these authors observed regeneration of the normal epithelial lining and submucosal fibrosis, particularly when larger balloon sizes had been used. We speculated that submucosal fibrosis after ETBD may stiffen and keep the ET patent. Another finding in the submucosa was increased vascularity which could be attributed to the healing process after mucosal damage caused by ETBD [30, 31].

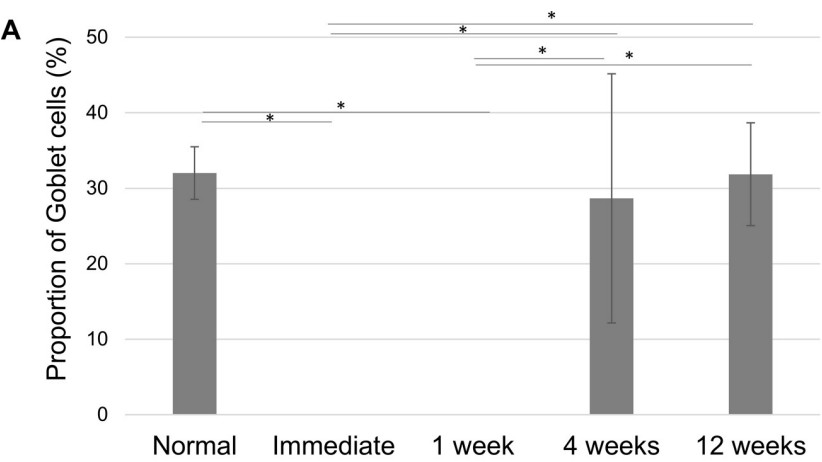

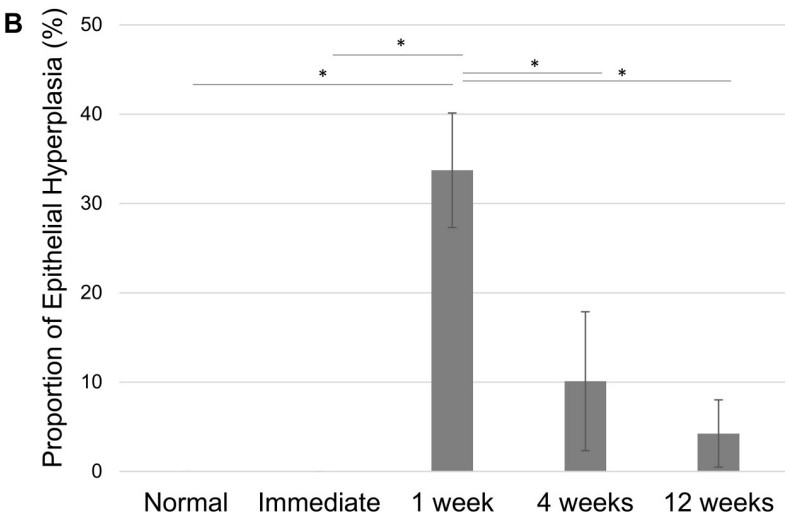

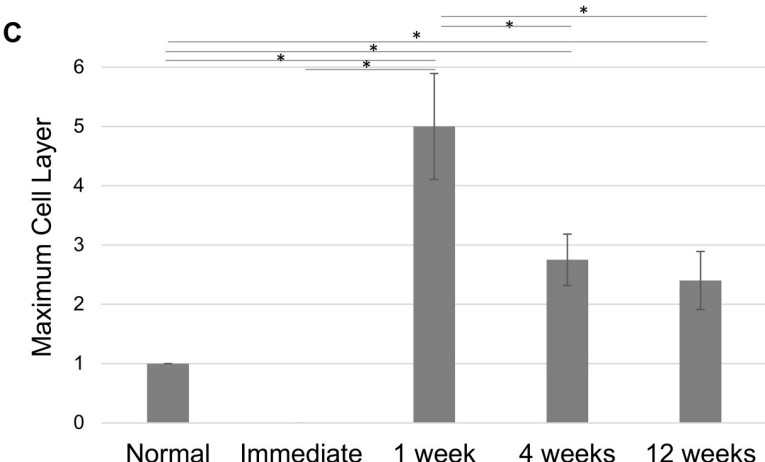

**Fig 5.** (A) Serial changes in the proportion of the goblet cells in the epithelium of the nasopharyngeal end of the rat ET. (B) Serial changes of the proportion of epithelial hyperplasia. (C) Serial changes of the maximum cell layer counts present in the epithelium of the cartilaginous end of the ET. *statistically significant differences for p<0.05, Mann-Whitney U test.

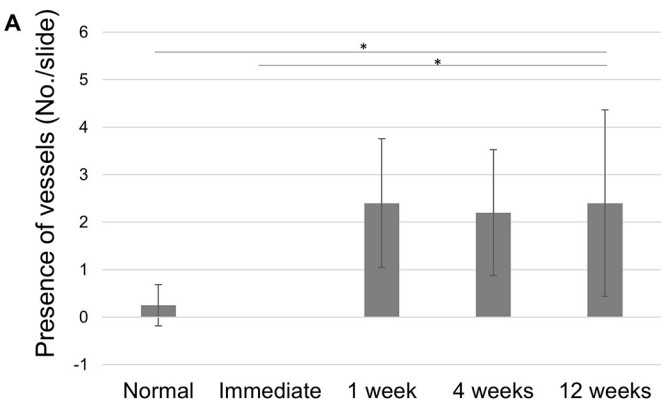

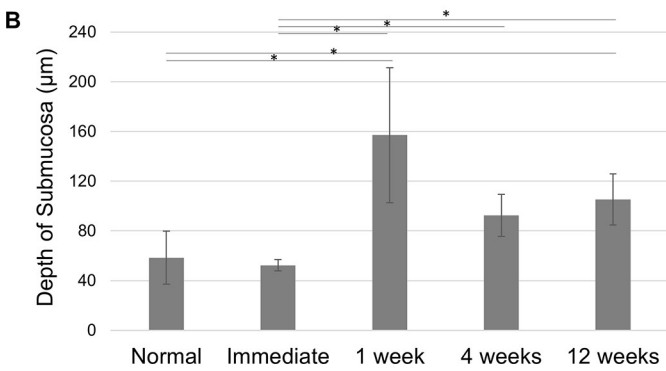

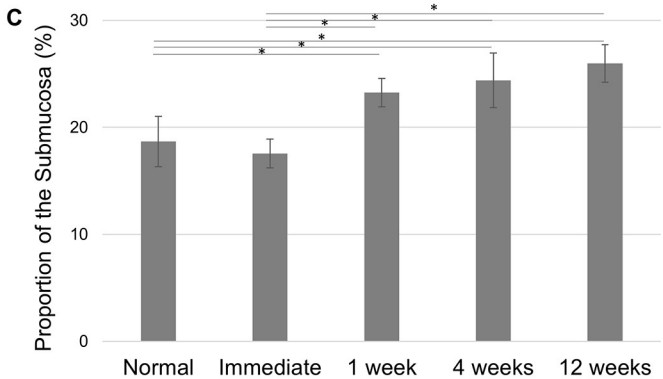

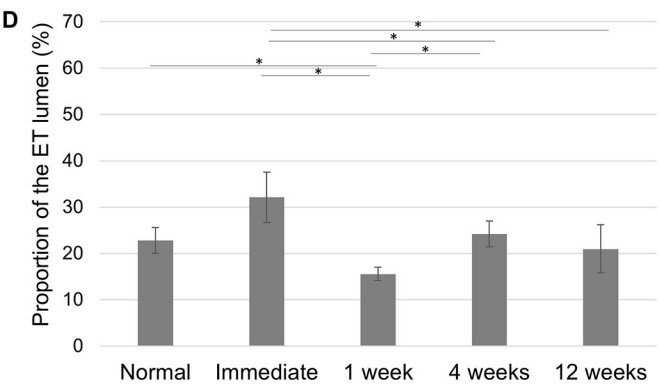

**Fig 6.** (A) Serial changes at the cartilaginous end of the rat ET including vascular structures; (B) submucosal depth; (C) the area of the submucosa (%) and (D) the area of the ET lumen (%). **statistically significant differences for p<0.05, Mann–Whitney U test.

In our present study of the rat ET, epithelial damage was observed immediately after ETBD. However, the mucosal lining regenerated at 1-week post-ETBD, the goblet cells had recovered by 4 weeks post-ETBD and the epithelial hyperplasia decreased to nearly normal levels at 12 weeks after the procedure. The epithelium thus seems to recover within 12 weeks. It can be argued that a regenerated healthy mucosa will be thinner, thus facilitating ET opening, and the possibly recovered function of the cilia and of mucus production may aid the clearance of secretions from the middle ear [32]. In addition, the ET mucosa is known to generate a surfactant that reduces surface tension, which would be reduced in otitis media with effusion [33, 34], and a restored healthy mucosa may secrete surfactant, thus aiding ET opening.

The dimensions of the tubal cartilage in the rat ET did not change throughout our current study period. The cartilage healing process after fracture in general involves an equilibrium between the deposition of type I collagen (scar tissue) and expression of type II collagen (repair). Small full-thickness cartilage defects are mainly replaced by fibrocartilage, whereas partial-thickness defects heal through the deposition of fibrous scar tissue [35]. Full-thickness cartilage fractures were induced in our current rat model. Furthermore, cartilaginous structures tend to return to their original shape unless the deformation is maintained for several months [36]. These factors could account for the lack of changes to the cartilage framework following ETBD and this may suggest that this is a safe procedure. If the thickness of the

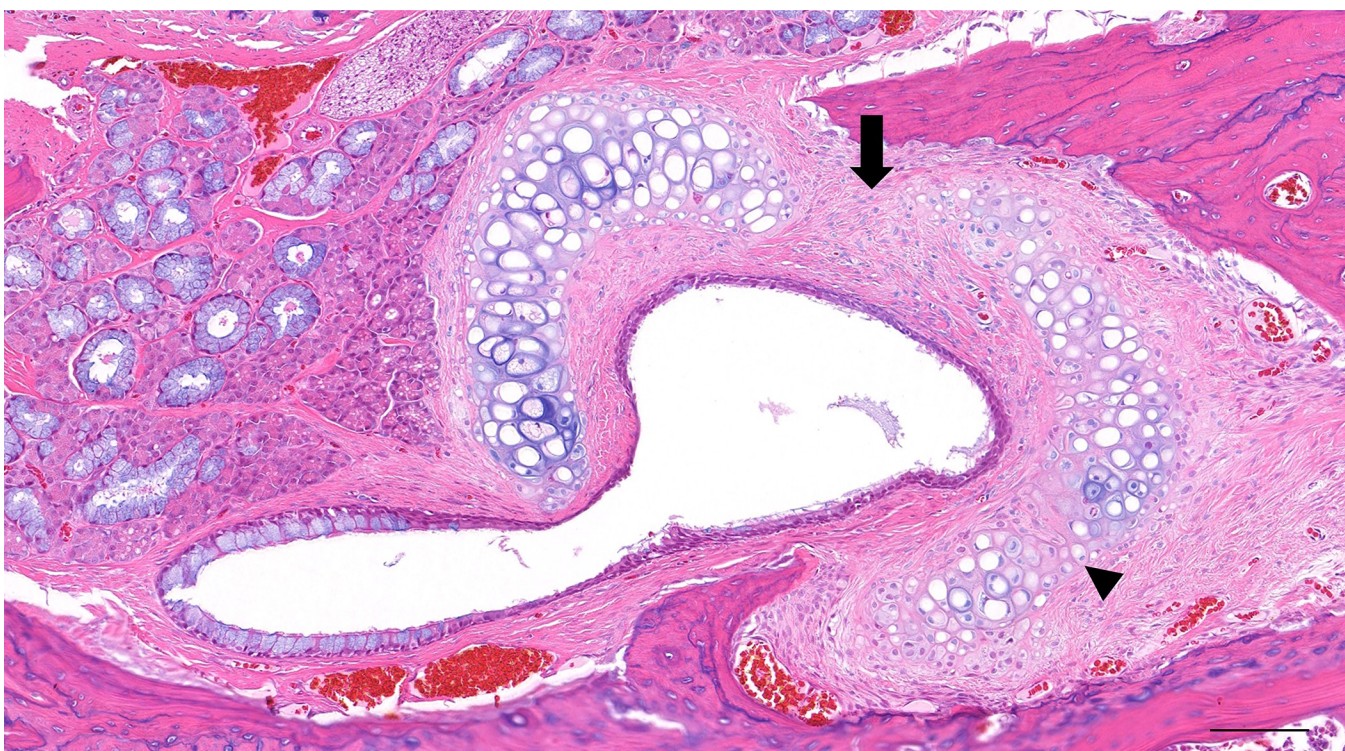

**Fig 7. Sites of cartilage fracture observed in 14 of the experimental rats.** The cartilage tended to fracture at three distinctive locations i.e. at the midpoint between the medial and lateral lamina of the tubal cartilage (arrow), and at the lateral two points that divide the lateral lamina into thirds (arrowhead). H&E, scale bar represents 100μm.

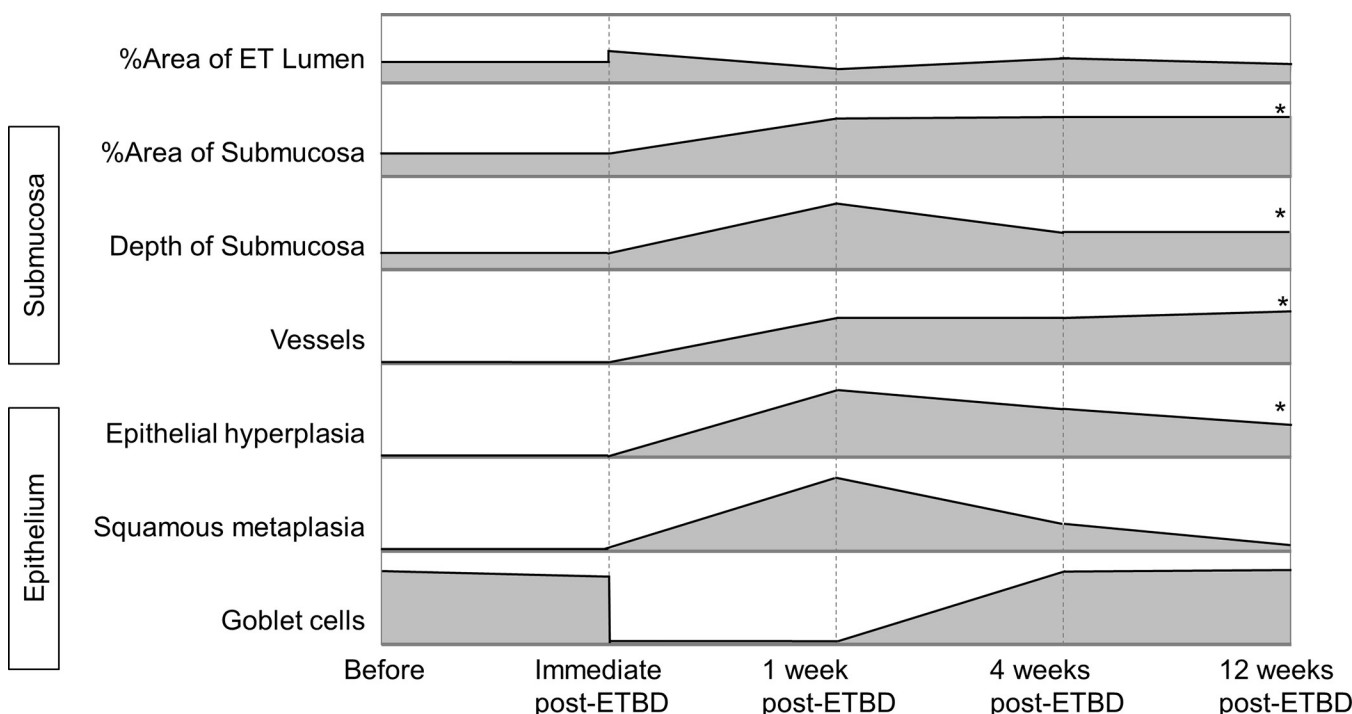

**Fig 8. Histologic changes of the lumen of the rat ET, the submucosa and the epithelium.** The asterisk denotes a significant difference between the period before surgery and at 12 weeks post-ETBD (p<0.05, Mann Whitney U test).

cartilage did change, we would not be able to exclude the possibility of cartilage contracture leading to further narrowing of the luminal stricture.

The main limitation of our current study was that no functional assessment of the ET could be done. The ET is a dynamic organ that remains closed at rest and opens during certain movements such as swallowing or yawning. The compliance of the ET is a known characteristic of its proper function [37]. Hence, we cannot yet ascertain how the submucosal fibrosis induced by ETBD affects the ET function. Another limitation was that our analyses were carried out on rats with a normal ET and we could not therefore analyze lymphocytic involvement. Moreover, ETBD was found to induce a thicker submucosa but no obvious changes in the lumen. In a chronically inflamed ET, the lumen would be narrower, and further studies in a rat model of otitis media and ETD might therefore provide better insights into the histologic changes caused by ETBD.

## Conclusion

Increased submucosal fibrosis and maintenance of the cartilage framework despite cartilage fractures after ETBD in the rat were the significant findings of our current study. At 12-weeks post-dilation, the observed findings were increased epithelial hyperplasia and increased depth of submucosa accompanied by neovascularization. This study is the first to describe serial histological changes after ETBD and these observations will assist with the planning of future histological studies of the middle ear in animal models after the placement of various types of balloons and stents that are typically used to treat intractable ET dysfunction in humans.

## Supporting information

**S1 Fig. Histology of the left-side rat ET sectioned immediately, at 1 week, 4weeks, and 12 weeks after ETBD.** Mucosal breaks are observed immediately after ETBD (arrow). Mucosal

wound is healed by 1 week and fibroblasts (asterisk) are seen. At 4 and 12 weeks, the submucosa is replaced with collagen tissue (arrowheads). Scale bars represent 100μm.
(TIF)

## Author Contributions

**Conceptualization:** Yehree Kim, Jung-Hoon Park, Hong Ju Park.

**Data curation:** Yehree Kim, Jeon Min Kang, Dae Sung Ryu, Jung-Hoon Park, Hong Ju Park.

**Formal analysis:** Yehree Kim, Hong Ju Park.

**Funding acquisition:** Hong Ju Park.

**Investigation:** Yehree Kim, Dae Sung Ryu, Jung-Hoon Park, Hong Ju Park.

**Methodology:** Yehree Kim, Jeon Min Kang, Hong Ju Park.

**Supervision:** Jung-Hoon Park, Hong Ju Park.

**Visualization:** Hong Ju Park.

**Writing – original draft:** Yehree Kim, Hong Ju Park.

**Writing – review & editing:** Jung-Hoon Park, Woo Seok Kang, Hong Ju Park.

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
