## [Decision Letter · Decision Letter 0]

1 Dec 2021

PONE-D-21-31219Serial Histological Changes in the Cartilaginous Eustachian Tube in the Rat Following Balloon DilationPLOS ONE

Dear Dr. Hong Ju Park,

Thank you for submitting your manuscript to PLOS ONE. After careful consideration, we feel that it has merit but does not fully meet PLOS ONE’s publication criteria as it currently stands. Therefore, we invite you to submit a revised version of the manuscript that addresses the points raised during the review process.

We look forward to receiving your revised manuscript.

Kind regards,

Ewa Tomaszewska, DVM Ph.D

Academic Editor

PLOS ONE

Journal Requirements:

2. To comply with PLOS ONE submissions requirements, in your Methods section, please provide additional information regarding the experiments involving animals and ensure you have included details on (1) methods to alleviate suffering, (2) methods of anesthesia and/or analgesia, and (3) basic housing and health monitoring

3. As part of your revision, please complete and submit a copy of the Full ARRIVE 2.0 Guidelines checklist, a document that aims to improve experimental reporting and reproducibility of animal studies for purposes of post-publication data analysis and reproducibility: https://arriveguidelines.org/sites/arrive/files/Author%20Checklist%20-%20Full.pdf (PDF). Please include your completed checklist as a Supporting Information file. Note that if your paper is accepted for publication, this checklist will be published as part of your article.

Reviewers' comments:

Reviewer's Responses to Questions

**Comments to the Author**

1. Is the manuscript technically sound, and do the data support the conclusions?

Reviewer #1: Partly

Reviewer #2: Partly

2. Has the statistical analysis been performed appropriately and rigorously? 

Reviewer #1: No

Reviewer #2: I Don't Know

3. Have the authors made all data underlying the findings in their manuscript fully available?

Reviewer #1: Yes

Reviewer #2: Yes

4. Is the manuscript presented in an intelligible fashion and written in standard English?

Reviewer #1: No

Reviewer #2: No

5. Review Comments to the Author

Reviewer #1: In the manuscript presented for the review the Authors describe the histological changes in the Eustachian tube (ET) after balloon dilation procedure (ETBD) at different intervals of time. I find the subject interesting and purposeful in the aspect of understanding processes that occur in the tissues after ETBD, which may result in the improvement of methodology and success rate of the procedure. A basic histological technique was used in this study to assess the epithelium, submucosa and the lumen of the ET. A series of morphometric analyses were also conducted to shed a light on the significance of the histological observations.

However, the manuscript was prepared carelessly and thus this article could be suitable for publication in PlosOne journal after major revisions.

Firstly, the Authors did not use the continuous line numbering which should be included according to a submission guideline and it would be helpful in preparing the review.

The English language should be improved throughout the whole manuscript. For example: “A proposed underlying mechanism of ETBD is microtears in the…” – a mechanism is the process. Microtears are not a mechanism. “The rats were randomly assigned to four groups of each five for serial…” - I guess there were 5 rats in each of four groups but the sentence is incorrect, “All rats were sacrificed immediately, 1 week, 4 weeks and 12 weeks.” – maybe adding “after ETBD” would make the sentence more understandable. “Normal Histological Findings for the Rat Eustachian Tube” – Were the histological findings normal or were they conducted in normal tissue?. “… worsening of the luminal stricture” – Can luminal stricture get worse? It is difficult to point out all the language discrepancies without line numbers.

ABSTRACT and INTRODUCTION. The aim of the study should be clearly pointed and precise. The statement that some issues “remain unclear” is insufficient and does not include the purpose of the study. The last sentence in the “introduction” should include not only the histological observations but also the morphometry and time intervals in which the alterations were observed.

MATERIALS AND METHODS. In “Animals” the sentences referring to conditions of animal housing and adaptation should be put before the description of the study. The method of anaesthesia should be described as it was done in the “Fluoroscopic Eustachian tube balloon dilation”. If the method is the same, it should be described when it appears in the text for the first time. Moreover, the Authors divide 20 animals into 4 groups but in “Histopathologic examination” section there are five groups mentioned with normal (before) group (probably from the right ear, but from which rats?). The division into study groups should be explained more precisely.

“Histopathologic examinations” should be changed to Histological examinations in accordance with the title of the manuscript. It should be also stated, how many slides were used from every animal for the morphological observations and how many microphotographs were taken for the morphometry (number of data taken into statistics). It should be explained how the depth of the submucosa relates to the fibrosis. Changes in the dimensions of the submucosa may be caused by different factors. Histopathological lesions should be marked on the figures if there were such alterations. Then they may be analysed morphometrically. In this case, the presence of fibrosis is a speculation.

“Statistical analysis”. The Mann Whitney U test is a nonparametric test. It should be explained why such test was used for the analyses. It should be mentioned if all the data did not meet the assumptions for the parametric test? For example normality of the distribution or homogeneity of variance. “A p-value of < 0.05 was considered statistically significant.” Is a p-value significant or the differences between groups? (Another unfortunate language expression).

RESULTS. If the Authors decide to emphasize the significance of the changes by adding “(p<0,05)” at the end of sentence they should follow this rule everywhere and not only in selected places. After some of the sentences, the Authors put “p=0,03”… does it mean that the result is ambiguous with p-value close to 0,05? After p-value, the name of the test should also be mentioned (also in legends to figures).

In “Serial changes in the tubal cartilage after ETBD” the Authors stated that a fracture lines were identified in 14 rats’ ET. After that the Authors write: “All the cartilage fracture lines were observed immediately after ETBD” which gives only 5 rats (immediately group) and not 14. It should be explained more precisely.

DISCUSSION. How the paragraph “In the urethra,…” relates to the changes in ET?

REFERENCES. The names of the Journals should be abbreviated in references: 24, 25, 29, 34, 37 and 41.

FIGURES AND FIGURE LEGENDS. The figures should contain markers of different structures of interest explained in the legends e.g. ET lumen, epithelium, Goblet cells, blood vessels, cartilage etc. for the readers who are not familiar with histology. The values on the scale bars in Fig 1 A,B,C and Fig 6 are not visible. The Authors should decide if they put the values on the scale bar or in the legends and follow their rule in all figures. In the Legends to figures, the Authors should avoid descriptions which are intended to be in the Results section. The H&E abbreviation should be used in all figures instead of full names. The letters “A,B,C” should be put at the beginning or at the end of the particular legend (the same arrangement in all figures). In the figure 7, the order of the parameters should be changed according to the legend to figure 7. The significant changes in the submucosa should be marked on the slides (e.g. fibrotic changes).

Another minor comments. After “Seongnam, Korea” there are 2 parentheses. “Ad libitum” should be in italics. In every “Fig.” there should be no dot. Abbreviation “BD” was not explained in the text. There was only ETBD.

Reviewer #2: The subject of this manuscript is interesting, as well as the obtained results. Unfortunately, the manuscript is not well written. Especially the methodological part requires major improvements. The manuscript cannot be published in this form.

Material and Methods

-It is said: In order to quantify changes in any of these aspects of the epithelium, the length of the epithelium exhibiting goblet cells (why the number of cells was not calculated?), and the level ( what does level mean – area, depth or length?) of squamous metaplasia or epithelial hyperplasia were measured and divided by the whole perimeter of the ET lumen in the corresponding slide (the charts in Figure 4 give percentages).

The authors did not clearly describe the methodology for quantifying these data. It is not clear to the reader how the authors made these calculations.

- The authors should use -blood vessels rather than vessels

- other staining should be used to identify fibrosis in the submucosa, e.g. Masson-Goldner's trichrome staining - This method would clearly visualize collagen fibers in the tissue

In the statistical analysis section there is no information on the presentation of data (as mean± SEM or SD?)

Results

It is said: the epithelial cells at the nasopharyngeal end were columnar and contained goblet cells. The epithelial cells at the cartilaginous end were more cuboidal in shape and contained few secretory mucous cells - These sentences are imprecise as its suggest that the epithelial cells contained secretory cells (the same applies to the description of Figure 1). Rather it should be: for example, the cartilaginous end epithelium consists of cuboidal and secretory cells.

- The authors write: The epithelial cells at the nasopharyngeal end of the ET rat were desquamated immediately after ETBD. It should be written - they were destroyed or damaged.

- It is said: At 4 weeks post-procedure, the epithelial hyperplasia decreased, and the original morphology was finally recovered at 12 weeks post-ETBD (Fig. 3). But in the earlier description, the authors did not mention anything about hyperplasia (here it appears for the first time, so it is difficult to understand when the authors write that the hyperplasia has decreased).

- The higher magnifications of the photos in Figure 2 are not sharp

- The authors should mark in Figure 3 the places where squamous metaplasia and hyperplasia occur.

- the data shown in the graphs (especially Figures 4 and 5A) have large SEM or standard deviations??- which proves the large dispersion of data

Discussion

-Based on the analysis of the depth of the submucosa, the authors conclude that there was an increase in fibrosis, however, the authors did not perform any analyzes confirming the fibrosis. It is worth using other staining that will confirm the fibrosis (e.g. the mentioned Masson-Goldner staining).

-the authors did not discuss the observed increase in vascularity in the submucosa. How to explain increased angiogenesis?

- Linguistic proofreading needed!!

6. PLOS authors have the option to publish the peer review history of their article (what does this mean?). If published, this will include your full peer review and any attached files.

Reviewer #1: No

Reviewer #2: No

---

## [Author Response · Author response to Decision Letter 0]

2 Feb 2022

We appreciate all the reviewers’ valuable and insightful comments that potentially improve our manuscript. We made our best effort to reflect the suggestions in the revised manuscript as well as to address all the concerns that the reviewers presented. Below are our line-by-line responses to the reviewers’ comments. 

Reviewer #1: In the manuscript presented for the review the Authors describe the histological changes in the Eustachian tube (ET) after balloon dilation procedure (ETBD) at different intervals of time. I find the subject interesting and purposeful in the aspect of understanding processes that occur in the tissues after ETBD, which may result in the improvement of methodology and success rate of the procedure. A basic histological technique was used in this study to assess the epithelium, submucosa and the lumen of the ET. A series of morphometric analyses were also conducted to shed a light on the significance of the histological observations.

However, the manuscript was prepared carelessly and thus this article could be suitable for publication in PlosOne journal after major revisions.

Firstly, the Authors did not use the continuous line numbering which should be included according to a submission guideline and it would be helpful in preparing the review.

→ Thankyou for pointing this out and we are sorry for omitting the numbering. We included the continuous line numbering in the revised manuscript. 

The English language should be improved throughout the whole manuscript. For example: “A proposed underlying mechanism of ETBD is microtears in the…” – a mechanism is the process. Microtears are not a mechanism. “The rats were randomly assigned to four groups of each five for serial…” - I guess there were 5 rats in each of four groups but the sentence is incorrect, “All rats were sacrificed immediately, 1 week, 4 weeks and 12 weeks.” – maybe adding “after ETBD” would make the sentence more understandable. “Normal Histological Findings for the Rat Eustachian Tube” – Were the histological findings normal or were they conducted in normal tissue?. “… worsening of the luminal stricture” – Can luminal stricture get worse? It is difficult to point out all the language discrepancies without line numbers.

→ Thankyou for the comments. “A proposed underlying mechanism of ETBD” was changed to “A reported finding after ETBD”. “The rats were randomly assigned to four groups of each five for serial…” was changed to “The rats were randomly assigned to 4 groups, each consisting of 5 animals, for serial histological examination at 4 different time points: immediately (n = 5), at 1 week (n = 5), 4 weeks (n = 5), and 12 weeks (n = 5) after ETBD”. “Normal Histological Findings for the Rat Eustachian Tube” was changed to “Histological Findings of Normal Rat Eustachian Tubes”. ““… worsening of the luminal stricture” was changed to ““… further narrowing of the luminal stricture””

ABSTRACT and INTRODUCTION. The aim of the study should be clearly pointed and precise. The statement that some issues “remain unclear” is insufficient and does not include the purpose of the study. The last sentence in the “introduction” should include not only the histological observations but also the morphometry and time intervals in which the alterations were observed.

→ Thankyou for the comments. The first 2 sentences in the ABSTRACT were changed to “Although balloon dilation has shown promising results in the treatment of dilatory Eustachian tube (ET) dysfunction, the histological effects of ET balloon dilation (ETBD) is unknown because histological examination of the whole human cartilaginous ET is impossible. Animal studies are needed to elucidate the effect of ETBD so we evaluated the histological changes after ETBD in a rat model.” The last sentence in the “introduction” was changed to “In this study, we aimed to evaluate the serial histological changes of the ET epithelium, submucosa, and cartilage and the area of the ET lumen immediately after and at 1, 4 and 12 weeks after balloon dilation in a rat model”

MATERIALS AND METHODS. In “Animals” the sentences referring to conditions of animal housing and adaptation should be put before the description of the study. The method of anaesthesia should be described as it was done in the “Fluoroscopic Eustachian tube balloon dilation”. If the method is the same, it should be described when it appears in the text for the first time. Moreover, the Authors divide 20 animals into 4 groups but in “Histopathologic examination” section there are five groups mentioned with normal (before) group (probably from the right ear, but from which rats?). The division into study groups should be explained more precisely.

→ Thankyou for the comments. The sentences referring to conditions of animal housing and adaptation were brought before the description of the study in the manuscript. Method of anesthesia was explained in the “Animals” section. Descriptions of the 4 groups and the how the right ear were used as normal control ear were rephrased for clearer communication. The revised version now reads “The left ET was dilated with a balloon catheter and the right ET was used as a normal control in all 20 rats. The rats were randomly assigned to 4 groups, each consisting of 5 animals, for serial histological examination at 4 different time points: immediately (n = 5), at 1 week (n = 5), 4 weeks (n = 5), and 12 weeks (n = 5) after ETBD.” and “These measurements were then compared among the 4 study groups (immediate, and 1, 4, and 12 weeks after ETBD) and the normal group (20 specimens from the non-dilated right ET”. 

“Histopathologic examinations” should be changed to Histological examinations in accordance with the title of the manuscript. It should be also stated, how many slides were used from every animal for the morphological observations and how many microphotographs were taken for the morphometry (number of data taken into statistics). It should be explained how the depth of the submucosa relates to the fibrosis. Changes in the dimensions of the submucosa may be caused by different factors. Histopathological lesions should be marked on the figures if there were such alterations. Then they may be analysed morphometrically. In this case, the presence of fibrosis is a speculation.

→ “Histopathologic examinations” was changed to “Histological examinations”. We made 7 slides per animal however one slide where ET cartilage was in the form of a ‘comma’ shape, showing both cartilaginous and nasopharyngeal epithelia was used for analysis. As provided in supplementary Figure 1, all 7 slides were reviewed and the section with the ‘comma’ shaped cartilage (slide with asterisk) was selected. How the depth of the submucosa relates to the fibrosis was also added “After ETBD, the mucosal breaks caused by the dilatory force of the balloon were replaced by collagen tissue. To evaluate the extent of fibrosis caused by ETBD, the depth of the submucosa which would correspond to the synthesis of new collagen was measured” and added a supplementary figure (S1_Fig) to show the histologic changes corresponding to the replacement with collagen tissue.

“Statistical analysis”. The Mann Whitney U test is a nonparametric test. It should be explained why such test was used for the analyses. It should be mentioned if all the data did not meet the assumptions for the parametric test? For example normality of the distribution or homogeneity of variance. “A p-value of < 0.05 was considered statistically significant.” Is a p-value significant or the differences between groups? (Another unfortunate language expression).

→ The whole “Statistical analysis” section was rewritten as “Statistical analyses were performed using SPSS software (version 24.0; SPSS, IBM, Chicago, IL). When comparing between two samples, significance of data was assessed by Mann-Whitney test because samples did not pass the normality test. A p-value of less than 0.05 was considered as statistically significant for differences between groups.”

RESULTS. If the Authors decide to emphasize the significance of the changes by adding “(p<0,05)” at the end of sentence they should follow this rule everywhere and not only in selected places. After some of the sentences, the Authors put “p=0,03”… does it mean that the result is ambiguous with p-value close to 0,05? After p-value, the name of the test should also be mentioned (also in legends to figures).

→ Thankyou for the comment. All the p-values were written as p<0.05 and the name of the test was written both in the results section and the legends to figures. 

In “Serial changes in the tubal cartilage after ETBD” the Authors stated that a fracture lines were identified in 14 rats’ ET. After that the Authors write: “All the cartilage fracture lines were observed immediately after ETBD” which gives only 5 rats (immediately group) and not 14. It should be explained more precisely.

→ We have erased the sentence as it may not be necessary and only cause confusion.

DISCUSSION. How the paragraph “In the urethra,…” relates to the changes in ET?

→ Our original intention was to discuss other examples of balloon dilation in non-vascular structures but decided to erase the paragraph as it may be irrelevant. Thankyou.

REFERENCES. The names of the Journals should be abbreviated in references: 24, 25, 29, 34, 37 and 41.

→ Thankyou for pointing this out. The correct journal abbreviations were used in the revised manuscript.

FIGURES AND FIGURE LEGENDS. The figures should contain markers of different structures of interest explained in the legends e.g. ET lumen, epithelium, Goblet cells, blood vessels, cartilage etc. for the readers who are not familiar with histology. The values on the scale bars in Fig 1 A,B,C and Fig 6 are not visible. The Authors should decide if they put the values on the scale bar or in the legends and follow their rule in all figures. In the Legends to figures, the Authors should avoid descriptions which are intended to be in the Results section. The H&E abbreviation should be used in all figures instead of full names. The letters “A,B,C” should be put at the beginning or at the end of the particular legend (the same arrangement in all figures). In the figure 7, the order of the parameters should be changed according to the legend to figure 7. The significant changes in the submucosa should be marked on the slides (e.g. fibrotic changes).

→ Thankyou for the comments. A new Fig1 (previously submitted as supplementary figure 1) was shown explaining the low power field view of the ET lumen and the cartilage at the materials&methods section. The epithelium, Goblet cells, blood vessels were marked in the new figures 3 and 4. The values of the scale bars were put in the legends and not the figures themselves. H&E abbreviations were used throughout all figures. The letters “A,B,C” were put at the beginning in all figure legends. The order of the legends of the new Fig8 (previous Fig7) were changed to fit the order of the figure.

Another minor comments. After “Seongnam, Korea” there are 2 parentheses. “Ad libitum” should be in italics. In every “Fig.” there should be no dot. Abbreviation “BD” was not explained in the text. There was only ETBD.

→ Thankyou for the comments. One of the 2 parentheses was erased. ‘Ad libitum’ was changed to italics. The dot after the ‘Fig.” were all removed. BD was changed to ETBD.

Reviewer #2: The subject of this manuscript is interesting, as well as the obtained results. Unfortunately, the manuscript is not well written. Especially the methodological part requires major improvements. The manuscript cannot be published in this form.

Material and Methods

-It is said: In order to quantify changes in any of these aspects of the epithelium, the length of the epithelium exhibiting goblet cells (why the number of cells was not calculated?), and the level ( what does level mean – area, depth or length?) of squamous metaplasia or epithelial hyperplasia were measured and divided by the whole perimeter of the ET lumen in the corresponding slide (the charts in Figure 4 give percentages). The authors did not clearly describe the methodology for quantifying these data. It is not clear to the reader how the authors made these calculations.

→ Thankyou for the comment. Because there may be some variations in the angles at which the ETs were sectioned (although we did try to keep the sections as coronal as possible checking with the hard palate and brain which were large enough to be grossly observed), we thought that proportions would better represent the changes rather than absolute numbers and counts. The sentences were rephrased and the revised manuscript now reads “In the epithelium, the presence of goblet cells, squamous metaplasia and epithelial hyperplasia was assessed. In order to quantify the changes of these aspects of the epithelium, the length of the epithelium exhibiting goblet cells, squamous metaplasia or epithelial hyperplasia were measured and divided by the whole perimeter of the ET lumen in the corresponding slide. Epithelial hyperplasia was also assessed by counting the maximum number of cell layers. These measurements were then compared among the 4 study groups (immediate, and 1, 4, and 12 weeks after ETBD) and the normal group (20 specimens from the non-dilated right ET).”

- The authors should use -blood vessels rather than vessels

→ All “vessels” were changed to “blood vessels”.

- other staining should be used to identify fibrosis in the submucosa, e.g. Masson-Goldner's trichrome staining - This method would clearly visualize collagen fibers in the tissue

→ Thankyou for the kind suggestion. In this study, we wanted to outline an overview of serial histologic changes after ETBD. Unfortunately, special stains were not considered when we were planning the experiments and only H&E sections were done. So collagen fibers had to be identified through eosin stain by looking at its morphology. In future studies, we plan to include special stains.

In the statistical analysis section there is no information on the presentation of data (as mean± SEM or SD?)

→ The sentence “The data are presented as means ± standard deviations” were added to the beginning of the statistical analysis section

Results

It is said: the epithelial cells at the nasopharyngeal end were columnar and contained goblet cells. The epithelial cells at the cartilaginous end were more cuboidal in shape and contained few secretory mucous cells - These sentences are imprecise as its suggest that the epithelial cells contained secretory cells (the same applies to the description of Figure 1). Rather it should be: for example, the cartilaginous end epithelium consists of cuboidal and secretory cells.

→ Thankyou for your kind suggestion. The sentences were rewritten as “The epithelium at the nasopharyngeal end consisted of columnar cells and also contained goblet cells (secretory mucous cells). The epithelium at the cartilaginous end consisted of cuboidal cells and contained few secretory mucous cells.”

- The authors write: The epithelial cells at the nasopharyngeal end of the ET rat were desquamated immediately after ETBD. It should be written - they were destroyed or damaged.

→ The word “desquamated” was changed to “destroyed” as suggested.

- It is said: At 4 weeks post-procedure, the epithelial hyperplasia decreased, and the original morphology was finally recovered at 12 weeks post-ETBD (Fig. 3). But in the earlier description, the authors did not mention anything about hyperplasia (here it appears for the first time, so it is difficult to understand when the authors write that the hyperplasia has decreased).

→ The term “epithelial hyperplasia” was erased and changed to “squamous metaplasia”.

- The higher magnifications of the photos in Figure 2 are not sharp

→ The figures were recaptured (759dpi) for better resolution

- The authors should mark in Figure 3 the places where squamous metaplasia and hyperplasia occur.

→ Thankyou for your suggestion. Squamous metaplasia and hyperplasia as well as blood vessels and Goblet cells etc were marked in the figures which was also suggested by reviewer 1. 

- the data shown in the graphs (especially Figures 4 and 5A) have large SEM or standard deviations??- which proves the large dispersion of data

→ For some variables (especially where absolute counts of small numbers were used) we did see large variance among animals.

Discussion

-Based on the analysis of the depth of the submucosa, the authors conclude that there was an increase in fibrosis, however, the authors did not perform any analyzes confirming the fibrosis. It is worth using other staining that will confirm the fibrosis (e.g. the mentioned Masson-Goldner staining).

→ Thankyou for the suggestion. Again, in this particular study, only H&E sections were done. We tried to include explanations for measuring the depth of the submucosa for representation of newly formed collagen in the material and methods section under “histologic examinations”. “After ETBD, the mucosal breaks caused by the dilatory force of the balloon were replaced by collagen tissue. To evaluate the extent of fibrosis caused by ETBD, the depth of the submucosa which would correspond to the synthesis of new collagen was measured” and added supplementary Fig 2 to show the histologic changes corresponding to the replacement with collagen tissue.”

-the authors did not discuss the observed increase in vascularity in the submucosa. How to explain increased angiogenesis?

→ Increased vascularity was discussed in the 4th paragraph of the Discussion section. “Another finding in the submucosa was increased vascularity which could be attributed to the healing process after mucosal damage caused by ETBD [31,32].”

- Linguistic proofreading needed!!

→ Thankyou for all your considerate suggestions. Certificate of language editing is included.

---

## [Decision Letter · Decision Letter 1]

15 Mar 2022

PONE-D-21-31219R1Serial histological changes in the cartilaginous Eustachian tube in the rat following balloon dilationPLOS ONE

Dear Dr. Hong Ju Park,

Thank you for submitting your manuscript to PLOS ONE. After careful consideration, we feel that it has merit but does not fully meet PLOS ONE’s publication criteria as it currently stands. Therefore, we invite you to submit a revised version of the manuscript that addresses the points raised during the review process.

We look forward to receiving your revised manuscript.

Kind regards,

Ewa Tomaszewska, DVM Ph.D

Academic Editor

PLOS ONE

Journal Requirements:

Reviewers' comments:

Reviewer's Responses to Questions

**Comments to the Author**

1. If the authors have adequately addressed your comments raised in a previous round of review and you feel that this manuscript is now acceptable for publication, you may indicate that here to bypass the “Comments to the Author” section, enter your conflict of interest statement in the “Confidential to Editor” section, and submit your "Accept" recommendation.

Reviewer #1: (No Response)

Reviewer #2: All comments have been addressed

2. Is the manuscript technically sound, and do the data support the conclusions?

Reviewer #1: Yes

Reviewer #2: Yes

3. Has the statistical analysis been performed appropriately and rigorously? 

Reviewer #1: Yes

Reviewer #2: Yes

4. Have the authors made all data underlying the findings in their manuscript fully available?

Reviewer #1: Yes

Reviewer #2: Yes

5. Is the manuscript presented in an intelligible fashion and written in standard English?

Reviewer #1: Yes

Reviewer #2: Yes

6. Review Comments to the Author

Reviewer #1: The manuscript entitled: “Serial histological changes in the cartilaginous Eustachian tube in the rat following balloon dilation” is interesting and purposeful in the aspect of understanding processes that occur in the tissues after Eustachian tube ballon dilation. The Authors have assessed the epithelium, submucosa and the lumen of the ET using a basic histological technique enriched with morphometric analyses.

The Authors have put a great effort to improve their manuscript after first revision. My comments and suggestions were taken into consideration, however there are still some minor issues that should be revised:

1. In References section, literature positions 21 and 24 are the same. After deleting one of them the numbers in the text should be corrected.

2. In References some names of journals are still not abbreviated:

Reference 25: J Vasc Interv Radiol

Reference 29: J Pharm Bioallied Sci

Reference 34: Int J Pediatr Otorhinolaryngol

Reference 38: Am J Otolaryngol

3. Figures and Figure legends

In Fig 3 and 4 the white arrowheads are barely visible

After * in fig 5 and 6 there should be: *statistically significant differences for p<0.05, Mann-Whitney U test.

In S1 Fig. there should be a scale bar or magnification indicated in the legend.

In my opinion, the manuscript is suitable for publication in PLOS ONE after minor revision.

Reviewer #2: The work has been corrected in accordance with the comments of the reviewer, but I do have some remark, however. I see ambiguity in the description of Figure 4 (lines 457-460) and the description of the Results (lines 190-192) - what exactly did the authors want to write - epithelial hyperplasia or squamous metaplasia?

7. PLOS authors have the option to publish the peer review history of their article (what does this mean?). If published, this will include your full peer review and any attached files.

Reviewer #1: No

Reviewer #2: No

---

## [Author Response · Author response to Decision Letter 1]

28 Apr 2022

We appreciate the reviewers’ valuable comments that potentially improve our manuscript. We made our best effort to reflect the suggestions in the revised manuscript. Below in blue are our line-by-line responses to the reviewers’ comments. 

Reviewer #1: The manuscript entitled: “Serial histological changes in the cartilaginous Eustachian tube in the rat following balloon dilation” is interesting and purposeful in the aspect of understanding processes that occur in the tissues after Eustachian tube ballon dilation. The Authors have assessed the epithelium, submucosa and the lumen of the ET using a basic histological technique enriched with morphometric analyses.

The Authors have put a great effort to improve their manuscript after first revision. My comments and suggestions were taken into consideration, however there are still some minor issues that should be revised:

1. In References section, literature positions 21 and 24 are the same. After deleting one of them the numbers in the text should be corrected.

→ Thank you for pointing this out. Reference 24 was removed and changed to 21.

2. In References some names of journals are still not abbreviated:

Reference 25: J Vasc Interv Radiol

Reference 29: J Pharm Bioallied Sci

Reference 34: Int J Pediatr Otorhinolaryngol

Reference 38: Am J Otolaryngol

→ Thank you for pointing this out. The abbreviations were corrected as suggested.

3. Figures and Figure legends

In Fig 3 and 4 the white arrowheads are barely visible

→ For Fig 4, the white arrowheads were changed to “white arrowheads with black borders” for better visualization. For Fig 3, there was a mistake with the legends so the sentence reading “White arrowheads denote blood vessels” was removed from the manuscript.

After * in fig 5 and 6 there should be: *statistically significant differences for p<0.05, Mann-Whitney U test.

→ Thank you for the suggestion. The suggested corrections were made to the manuscript.

In S1 Fig. there should be a scale bar or magnification indicated in the legend.

→ Scale bars were added to the figure and indicated in the legend

In my opinion, the manuscript is suitable for publication in PLOS ONE after minor revision.

→ Thank you for your positive feedback.

Reviewer #2: The work has been corrected in accordance with the comments of the reviewer, but I do have some remark, however. I see ambiguity in the description of Figure 4 (lines 457-460) and the description of the Results (lines 190-192) - what exactly did the authors want to write - epithelial hyperplasia or squamous metaplasia?

→ There was confusion with the choice of terminology in the original manuscript. Thank you for pointing this out. In the new manuscript, the finding was uniformly described as epithelial hyperplasia.

---

## [Decision Letter · Decision Letter 2]

9 May 2022

Serial histological changes in the cartilaginous Eustachian tube in the rat following balloon dilation

PONE-D-21-31219R2

Dear Dr. Hong Ju Park,

We’re pleased to inform you that your manuscript has been judged scientifically suitable for publication and will be formally accepted for publication once it meets all outstanding technical requirements.

Kind regards,

Ewa Tomaszewska, DVM Ph.D

Academic Editor

PLOS ONE

Reviewers' comments:

Reviewer's Responses to Questions

**Comments to the Author**

1. If the authors have adequately addressed your comments raised in a previous round of review and you feel that this manuscript is now acceptable for publication, you may indicate that here to bypass the “Comments to the Author” section, enter your conflict of interest statement in the “Confidential to Editor” section, and submit your "Accept" recommendation.

Reviewer #2: All comments have been addressed

2. Is the manuscript technically sound, and do the data support the conclusions?

Reviewer #2: Yes

3. Has the statistical analysis been performed appropriately and rigorously? 

Reviewer #2: Yes

4. Have the authors made all data underlying the findings in their manuscript fully available?

Reviewer #2: Yes

5. Is the manuscript presented in an intelligible fashion and written in standard English?

Reviewer #2: Yes

6. Review Comments to the Author

Reviewer #2: The work has been corrected in accordance with the comments of the reviewer. This manuscript is accepted for publication.

7. PLOS authors have the option to publish the peer review history of their article (what does this mean?). If published, this will include your full peer review and any attached files.

Reviewer #2: No

---

## [Editor Report · Acceptance letter]

16 May 2022

PONE-D-21-31219R2 

Serial histological changes in the cartilaginous Eustachian tube in the rat following balloon dilation 

Dear Dr. Park:

I'm pleased to inform you that your manuscript has been deemed suitable for publication in PLOS ONE. Congratulations! Your manuscript is now with our production department. 

Kind regards, 

on behalf of

Professor Ewa Tomaszewska 

Academic Editor

PLOS ONE